# Evaluating Physician Adherence to Antithrombotic Recommendations in Patients with Atrial Fibrillation: A Pathway to Better Medical Education

**DOI:** 10.3390/ijerph17114008

**Published:** 2020-06-04

**Authors:** Ştefan Cristian Vesa, Sonia Irina Vlaicu, Octavia Sabin, Vitalie Văcăraș, Sorin Crișan, Sabina Istratoaie, Fatuma Samantar, Daciana Elena Popa, Antonia Eugenia Macarie, Anca Dana Buzoianu

**Affiliations:** 1Department of Pharmacology, Toxicology and Clinical Pharmacology, “Iuliu Haţieganu” University of Medicine and Pharmacy, 400337 Cluj-Napoca, Romania; stefanvesa@gmail.com (Ş.C.V.); sabina.istratoaie@gmail.com (S.I.); abuzoianu@umfcluj.ro (A.D.B.); 2Department of Internal Medicine, 1st Medical Clinic, “Iuliu Haţieganu” University of Medicine and Pharmacy, 400006 Cluj-Napoca, Romania; vlaicus@yahoo.com; 3Department of Neurology, “Iuliu Hațieganu” University of Medicine and Pharmacy, 400012 Cluj-Napoca, Romania; 4Department of Internal Medicine, 5th Medical Clinic, “Iuliu Haţieganu” University of Medicine and Pharmacy, 400139 Cluj-Napoca, Romania; crisan.sorin@gmail.com; 5Graduate of “Iuliu Haţieganu” Faculty of Medicine, University of Medicine and Pharmacy, 400337 Cluj-Napoca, Romania; fatumasam@yahoo.co.uk; 6Department of Cardiology, “Niculae Stăncioiu” Heart Institute, 400001 Cluj-Napoca, Romania; dacianaelenapopa@gmail.com; 7Department of Geriatrics-Gerontology, “Iuliu Haţieganu” University of Medicine and Pharmacy, 400139 Cluj-Napoca, Romania; macarieantonia@yahoo.com

**Keywords:** atrial fibrillation, stroke risk, adherence to anticoagulant treatment

## Abstract

*Background:* Atrial fibrillation is a major health problem due to the stroke risk associated with it. To reduce stroke risk, oral anticoagulants (OAC) are prescribed using the CHA_2_DS_2_-VASc (Congestive heart failure; Hypertension; Age ≥75 years; Diabetes Mellitus; Stroke; Vascular disease; Age 65–74 years; Sex category) risk score, a clinical probability assessment that includes a combination of risk factors predicting the probability of a stroke. Not all patients with high risk are receiving this treatment. The aim of this study was to assess physician adherence to clinical guidelines concerning the OAC treatment and to identify the factors that were associated with the decision to prescribe it. *Methods*: Registry data from 784 patients with non-valvular atrial fibrillation were evaluated in this retrospective cross-sectional study. Demographic data, subtype of AF, comorbidities associated with higher stroke and bleeding risk, and antithrombotic treatment received were recorded. We compared stroke and bleeding risk in patients with and without OAC treatment to determine if the clinicians followed guidelines: prescribed when necessary and abstained when not needed. *Results:* OAC treatment was administered in 617 (78.7%) patients. Of the 167 patients who did not receive OAC, 161 (96.4%) were undertreated according to their risk score, as opposed to those who received OAC in which the percentage of overtreated was 3.2%. Most undertreated patients (60.5%, *p* < 0.001) were with paroxysmal atrial fibrillation subtype. *Conclusions:* The decision to use anticoagulants for stroke prevention was based on the type of atrial fibrillation, rather than on the risk of stroke as quantified by CHA2DS2-VASc as per the recommended guidelines.

## 1. Introduction

Atrial fibrillation (AF) patients are four to five times more at risk of stroke than the general population and have a doubled mortality rate [1]. If the patient is correctly prescribed an oral anticoagulant (OAC) and the risk factors are effectively controlled, the risk for stroke decreases substantially and the patient lives longer [2]. In this regard, there is a broad consensus among international guidelines [2,3,4] for AF antithrombotic treatment toward the use of CHA_2_DS_2_-VASc (Congestive heart failure; Hypertension; Age ≥75 years; Diabetes Mellitus; Stroke; Vascular disease; Age 65–74 years; Sex category) score to assess stroke risk, considering the variables that have been shown to increase stroke risk, and validated in a large population. This score simplifies the clinical decision to prescribe OAC. Patients with a CHA_2_DS_2_-VASc score of 2 or more for men, and 3 or more for women, need OAC to reduce their stroke risk, while patients without clinical stroke risk factors do not benefit from antithrombotic therapy. The CHA_2_DS_2_-VASc acronym summarizes the risk factors taken into account for calculating the stroke risk: age, gender, congestive heart failure, arterial hypertension, stroke/transient ischemic attack/thromboembolism history, vascular disease history (prior myocardial infarction, peripheral artery disease or aortic plaque) and diabetes [5].

Despite this evidence and guideline recommendations, undertreatment with OACs is still common, due to fears or false perceptions in patients or doctors. The most widespread reason for withholding OAC is related to the calculated bleeding risk, bleeding events, or a perceived “high risk of bleeding”. Other reasons are in relation to the type of OAC medication, the efforts required to monitor the patients, the dose-adjusted vitamin K antagonists (VKA) therapy or the price for the non-vitamin K oral anticoagulants (NOACs) [2]. Some patients are at “high risk” calculated by bleeding risk scores and others are perceived as “higher risk” for bleeding: elderly, patients with cognitive dysfunction, frequent falls or frailty. However, even in those patients, the stroke risk without OAC often exceeds the bleeding risk [2,6,7,8]. 

Another error is the replacement of OAC with antiplatelets. Although antiplatelets are still perceived by some clinicians to be a safer alternative to OAC [9], there are numerous studies that prove that the bleeding risk on aspirin is no different to the bleeding risk on VKA or NOAC therapy, whereas VKA and NOACs effectively reduce strokes in AF patients while antiplatelets do not [2,9,10,11].

Prior to the introduction of the AF guidelines with stroke risk scores (initially CHADS_2_, then CHA_2_DS_2_-VASc) and the introduction of NOACs, the use of OAC (VKA) in the “real world” for the prevention of stroke was suboptimal, with only 15%–44% of patients with AF who had no contraindication to the therapy, actually receiving it [12]. Following those changes, an improved OAC prescription and a reduced risk of stroke was observed in Europe. The proportion of patients receiving OAC has increased, from ~50% to >80% of “high risk” AF patients, whereas the proportion of those treated with aspirin only or those left untreated has diminished [8,13,14,15]. 

However, there remains a significant percentage of patients that are not receiving the correct antithrombotic treatment—as low as 50% [16]. A high adherence to recommendations is associated with better outcomes in AF [13,15,17]. A study that used customized multilevel education intervention showed an increase in the use of OAC in patients with AF [18]. Because the intervention must be specific to each region, our study was designed to identify the actual factors associated with the underuse of the OAC, in order to improve the risk management.

For many years now, the controversy over stroke risk depending on the subtype of AF (paroxysmal, persistent or permanent) has been ongoing, due to divergent results in clinical trials [19,20,21]. As a consequence of these results, the current guidelines do not differentiate OAC recommendations between the subtypes of AF, and because there is no according risk stratification, OAC should be prescribed in high risk patients, irrespective of the AF pattern.

The aim of this study is to evaluate whether, in a regional hospital, OAC is prescribed to “high risk” stroke patients, whether antiplatelets are overused in patients, and to assess overall adherence to clinical guidelines concerning OAC treatment in AF. Secondly, we want to identify the factors that are associated with the decision to prescribe OAC. Evaluating physician adherence to antithrombotic recommendations is the first step on a pathway to better understanding the biases and restraints in following guidelines, and to setting up education programs to improve them.

## 2. Materials and Methods 

This retrospective, observational, cross-sectional study took place in an internal medicine and cardiology ward of a regional hospital in the city of Cluj-Napoca, Romania. We analyzed registry data collected from all patients admitted with a diagnosis of non-valvular AF in a time frame of 18 months from January 2017 until June 2018. The exclusion criteria were patients under the age of 18, patients with current active bleeding and patients who died during the admission period. The study was approved by the hospital’s ethics committee (no. 350/13 November 2014) and all patients signed an informed consent. 

In the demographic data, subtype of AF, comorbidities associated with higher stroke and bleeding risk and antithrombotic treatment received were recorded. All of the variables included in this study—namely congestive heart failure, hypertension, diabetes mellitus, previous stroke/transient ischemic attack (TIC) and vascular disease (prior MI, peripheral artery disease, or aortic plaque) for the CHA_2_DS_2-_VASc score; and liver disease (cirrhosis or Bilirubin > 2× normal or AST/ALT/AP > 3× normal), renal disease (dialysis, serum creatinine > 2.26 mg/dL), history of major bleeding, excessive alcohol use (>8 drinks/week) and concomitant use of antiplatelets for the HAS-BLED (Hypertension, Abnormal renal and liver function, Stroke, Bleeding, Labile INR, Elderly, Drugs or alcohol) score—were recorded during the admission of the patient as part of their medical history. We did not include labile INR in the HAS-BLED calculation, as a significant percentage of the patients followed NOAC therapy. The “high risk” for stroke was considered a CHA_2_DS_2_-VASc risk score of 2 or more in men, and 3 or more in women. 

Additionally, the variables were stratified and quantified in terms of OAC treatment. We compared stroke and bleeding risk in patients both with OAC treatment and without OAC treatment to determine if the clinicians followed ESC 2016 Atrial Fibrillation guidelines, by prescribing when necessary and abstaining when not needed. The ESC guideline was used as a standard of care in the hospital at the time of study. Patients were divided according to stroke risk, and antithrombotic treatment was indicated in undertreated if they were at “high risk” without OAC treatment, and as correctly treated or overtreated if an antiplatelet was added without other cardiovascular pathology justifying it.

A statistical analysis was carried out using the MedCalc Statistical Software version 19.1.5 (MedCalc Software bv, Ostend, Belgium, 2020). The normality of distribution for the quantitative data was verified with the Shapiro–Wilk test. The baseline data of patients were reported as median (25th, 75th percentiles) for continuous variables and frequencies and percentages for categorical variables. Between-group comparisons were performed using the Mann–Whitney test for continuous variables and chi-square tests or Fisher’s exact test, for categorical variables. A multivariate analysis was carried out using binary logistic regression. A *p* value < 0.05 was considered statistically significant.

## 3. Results

In the timeframe allocated for recruitment, we initially assessed a number of 835 patients for eligibility. Fifty-one patients were subsequently excluded from the initial number, due to the lack of available data. Our study enrolled 784 patients from which 167 patients (21.3%) had no OAC in their treatment and 617 patients had OAC; 488 (62.2%) of the 627 received VKAs and 129 (16.5%) received NOACs (Figure 1). 

Patient demographic and clinical data in relation to AC therapy are summarized in Table 1. 

The CHA_2_DS_2_-VASc score did not differ between patients with OAC and those without OAC. A HAS-BLED score ≥ 3 was associated with a low probability for the patient to receive OAC. The persistent or permanent AF was also associated with the presence of OAC. Of the total number of patients with a HAS-BLED score ≥ 3 (87 patients), 56 patients (64.36%) had at least one (60.7%) or two modifiable factors (39.3%): alcohol consumption (22 (39.3%), uncontrolled arterial hypertension (30 (53.6%)) or antiplatelet use without a recent history of MI (26 (46.4%)).

The frequency of AC medication was slightly higher in the group at higher risk of stroke. The difference was statistically insignificant (*p* = 0.1) (Table 2).

Patients with paroxysmal AF were less likely to receive OAC, even though they had a high risk of stroke and low risk of bleeding (Table 3). Patients with persistent/permanent AF were more likely to receive AC regardless of the risk of stroke and bleeding.

In order to evaluate the independent association between several parameters and the decision to administer oral anticoagulants, we used a multivariate logistic regression (Table 4). The variables that achieved significance in the univariate analysis were introduced into the model: the patient’s HAS-BLED score and the type of AF. A HAS-BLED score higher than 3 was associated with a low probability of AC treatment. Patients with persistent or permanent AF were more likely to receive AC treatment, compared to the ones with paroxysmal AF.

## 4. Discussion

Studies in several countries over the last two decades have consistently found a progressively steady increase in the proportion of AF patients with appropriately prescribed OACs [15,22]. The analysis of the prescribing habits of OAC and adherence to the current treatment guidelines is essential for assessing specific educational objectives in the healthcare provider’s medical education. The study results show the status quo regarding anticoagulation in a non-tertiary regional center. Despite being a single-center study, our research did enroll a large number of patients; the fact that others have (partially) reported similar findings supports the validity of our data. These results stand as a promontory in addressing the relevant specific issues regarding adherence of anticoagulation in AF patients. As such, our data could be used for targeted interventions, especially in medical units in countries with a matching socio-economic background.

More than 20% of the patients enrolled in our study who would have benefited from OAC treatment did not receive it. Non-adherence to guidelines is highly prevalent among elderly AF patients, despite guideline-adherent treatment being independently associated with lower risk of all-cause and CV deaths [8]. In one multicenter study in the Balkan region (including a center from Romania), age ≥ 80 years, prior myocardial infarction and paroxysmal AF were independent predictors of OAC non-use [2,14]. In our study, age was not a predictor of OAC non-use, and neither were myocardial infarction antecedents, but paroxysmal AF was found to be associated with a lower adherence to guidelines. These data are consistent with the results of other similar studies [8,14,22,23]. The proportion of patients with AF who were treated according to recommended guidelines increased by more than 50% over the decade from 2005–2015 for all stroke risk categories combined, but heterogeneously within and between countries, and OAC use overall remained sub-optimal [22]. 

A 2016 meta-analysis showed that persistent and permanent subtypes of AF entailed an increased risk of stroke as compared to paroxysmal AF [19], and one study showed a worse acute clinical course and greater volume of infarction in the case of permanent AF [24]. Therefore, the reluctance of clinicians to prescribe OAC in the case of paroxysmal fibrillation is partially justified, but this approach is hazardous as we do not have a clear criteria of stroke risk differentiation between these different subtypes. New biological markers—such as markers of endothelial function, pro-inflammatory molecules, or genetic polymorphisms [25], or other risk factors, such as subtype of AF, obstructive sleep apnea or obesity [2,19,26]—can be added to stroke risk calculation models for an improved assessment, but a different risk score will need validation in large studies. The CHA_2_DS_2_-VASc risk score continues to be the main tool. According to the 2016 ESC guidelines for AF, all subtypes of AF should be treated similarly in terms of anticoagulation therapy. Efforts to improve guideline adherence would lead to better outcomes for all AF patients, no matter the subtype of AF.

The HAS-BLED score can aid in identifying correctable bleeding risk factors, but it should not be used exclusively as a contraindication to starting anticoagulant therapy [2]. This statement is an encouragement to treat “high risk” stroke patients despite their HAS-BLED scores and to work alongside the patient to reduce their risk factors. The initiation of therapy should not be a passive one, but rather an active integrative approach between the clinician and the patient [2]. A frequent concern is related to the change in the quality of life of patients with OAC versus those without anticoagulants. Studies have shown that there are no significant differences in this regard, even in older patients [27,28].

Unfortunately, the aforementioned approach does not always coincide with the real-life practice. An important cause incriminated is the “fear of hemorrhage” derived from the clinician’s experience. Clinicians overestimate the potential risk of bleeding and underestimate the potential preventative stroke benefit. The OAC’s adverse effects are given a higher importance than initiating on a beneficial therapy plan. Multiple studies of AF patients and prescribing habits have demonstrated that the reasons for not initiating OACs are past bleeding episodes, the presence of anemia, and the perception among doctors of a lower stroke risk [22,23,29,30]. This demonstrates the overbearing weight of bleeding on the initiation of therapy. However, it does provide scope for the improvement of treatment, as education and regular emphasis on the importance of treating high-risk patients can override the ‘fear of hemorrhage’ experienced by the clinician.

In our study, we have shown that there was a statistical significance between patients who received and did not receive anticoagulants as a function of the AF type. According to the 2016 ESC guidelines for AF, all types of AF should be treated similarly in terms of anticoagulation therapy. According to studies based off of the National Cardiovascular Data registry in the USA or the National Health Service in the United Kingdom, clinicians were significantly less likely to prescribe oral anticoagulants to paroxysmal AF patients compared to those with permanent AF [22,31]. This falls in line with our finding that the type of AF was a determining factor in the decision to treat with OAC. A possible explanation resides in a false belief that the incidence of stroke is lower in paroxysmal AF than in permanent AF, a presumption which is not reflected by the AF guidelines. It is paramount to treat them both similarly, due to the similar risk levels of thromboembolic complications.

For more than 30% of our patients with a HAS-BLED score of > 3, risk factor management and patient education may translate into a shift towards a lower risk category. Patient education can improve self-care, long-term outcomes and adherence to therapy [2,17]; therefore, support tools based on clinical practice guidelines were developed by professional healthcare associations. Furthermore, online tools and mobile apps about AF and the risk of stroke have been created by independent parties (patient associations, doctors, pharmaceutical companies), but most of the AF apps are lacking scientific validation. The potential opportunity exists for a multi-disciplinary effort by regulatory agencies, healthcare organizations, and app developers to improve relevance and scientific validity [30]. More validated tools should be used to help both patients and physicians, because many aspects of assessment and communication in AF management are time consuming. 

Additionally, it is important to note that even in a highly specialized cardiology unit we did not observe the proper use of AF guidelines, although guideline adherence was better than in other studies [22]. This demonstrates that awareness and knowledge of the potential stroke risk and the varying scores available does not necessarily mean that clinicians are more predisposed to treat with OAC. An integrated approach to better the AF management (the ABC pathway) was proposed to improve the patient’s outcome. The “A” section of the program emphasizes the proper use of OACs, considering both the risk of stroke (CHA_2_DS_2_-VASc score) and of bleeding (HAS-BLED score). According to the proposed method, three steps should be observed: identification of low-risk patients that do not need anticoagulation, the offering of stroke protection for patients at risk and the choice of proper OAC (VKA or NOAC) [32]. This method was proved useful by Pastori et al., as they found a 49% reduction in cardiovascular events in the group that followed proper anticoagulation treatment [33]. This type of approach, combined with multilevel educational intervention on the use of oral anticoagulation, as described by Vinereanu et al. [18], can greatly improve the AF patient’s prognosis. 

Our research has some limitations: the study was both single-center and retrospective, and the prevalence of some variables was lower than in reality, due to the fact that they are not usually recorded.

## 5. Conclusions

The decision to use the OAC for stroke prevention was based on the type of AF, rather than on the risk of stroke as quantified by CHA_2_DS_2_-VASc (as per the recommended guidelines). A high HAS-BLED score was associated with a lower rate of OAC administration. Educational strategies that will improve the physician’s adherence to guidelines are needed, as more than 20% of patients with AF at risk of stroke do not receive the recommended OAC treatment.

## Figures and Tables

**Figure 1 ijerph-17-04008-f001:**
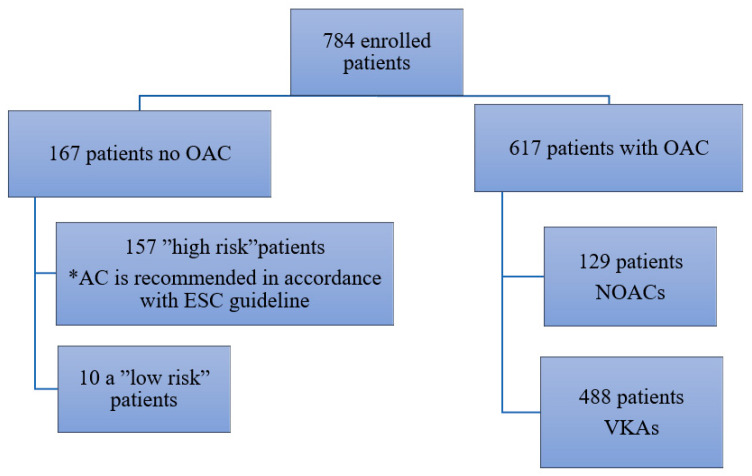
Patients distribution regarding the OAC treatment.

**Table 1 ijerph-17-04008-t001:** Patients characteristics in relation to AC therapy.

Variables		No AC (*n* = 167)	AC (*n* = 617)	*p*
Age (years) *		77 (69; 82)	76 (69; 82)	0.5
Age category	< 65 years	24 (14.4%)	102 (16.5%)	0.7
	65–75 years	52 (31.1%)	189 (30.6%)	
	> 75 years	91 (54.5%)	326 (52.8%)	
Gender	Male	76 (45.5%)	305 (49.4%)	0.3
	Female	91 (54.5%)	312 (50.6%)	
Heart failure		130 (77.8%)	507 (82.2%)	0.2
Arterial hypertension		131 (78.4%)	513 (83.1%	0.1
Diabetes mellitus		48 (28.7%)	170 (27.7%)	0.7
History of stroke/TIC		28 (16.8%)	109 (17.7%)	0.7
Vascular disease history		35 (21%)	124 (20.1%)	0.8
CHA_2_DS_2_-VASc *		4 (3; 5)	4 (3; 5)	0.9
CHA_2_DS_2_-VASc	No AC recommendation	10 (6%)	26 (4.2%)	0.4
	AC recommended	157 (94%)	591 (95.8%)	
Antiplatelet		76 (45.5%)	29 (4.7%)	< 0.001
Alcohol use		15 (9%)	71 (11.5%)	0.3
Prior major bleeding		9 (5.4%)	25 (4.1%)	0.4
Liver disease **		2 (1.2%)	8 (1.3%)	0.9
Renal diseases **		8 (4.8%)	30 (4.9%)	0.9
Uncontrolled arterial hypertension		18 (10.8%)	55 (8.9%)	0.4
HAS-BLED *		2 (1; 2)	1 (1; 2)	< 0.001
HAS-BLED	< 3	134 (80.2%)	563 (91.2%)	< 0.001
	≥ 3	33 (19.8%)	54 (8.8%)	
AF	Paroxysmal	101 (60.5%)	156 (25.3%)	< 0.001
	Persistent	6 (3.6%)	43 (7%)	
	Permanent	60 (35.9%)	418 (67.7%)	
Adherence to guideline	Undertreated	161 (96.4%)	-	< 0.001
	Correctly treated	6 (3.6%)	597 (96.8%)	
	Overtreated	-	20 (3.2%)	

* Median (25; 75 percentile); ** liver and renal disease are defined according to the HAS-BLED.

**Table 2 ijerph-17-04008-t002:** OAC and/or antiplatelet treatment according to the risk of stroke.

Variable	No AC Recommendation	AC Recommended
No antithrombotic treatment	8 (22.2%)	83 (11.1%)
Antiplatelet	2 (5.6%)	74 (9.9%)
OAC	26 (72.2%)	562 (75.1%)
Antiplatelet + OAC	-	29 (3.9%)

**Table 3 ijerph-17-04008-t003:** AC treatment regarding the type of AF, HASBLED score and CHA2DS2-VASc score.

	Paroxysmal AF				Persistent/Permanent AF			
	HAS-BLED				HAS-BLED			
	< 3		≥ 3		< 3		≥ 3	
No AC recommendation	24 (10.7%)		-		12 (2.5%)		-	
AC recommended	200 (89.3%)		33 (100%)		461 (97.5%)		54 (100%)	
	No AC	AC	No AC	AC	No AC	AC	No AC	AC
No AC recommendation	9 (11.1%)	15 (10.5%)	-	-	1 (1.9%)	11 (2.6%)	-	-
AC recommended	72 (88.9%)	128 (89.5%)	20 (100%)	13 (100%)	52 (98.1%)	409 (97.4%)	13 (100%)	41 (100%)

**Table 4 ijerph-17-04008-t004:** Multivariate analysis for the OAC treatment.

Variables	B	*p*	OR	95% C.I. for OR
Min	Max
HAS-BLED >3	−0.95	<0.001	0.38	0.23	0.63
Paroxysmal AF		-			
Persistent AF	1.48	0.001	4.42	1.80	10.83
Permanent AF	1.51	<0.001	4.55	3.13	6.61
Constant	1.08	<0.001	2.96

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
