# Peer review of "Evaluating Physician Adherence to Antithrombotic Recommendations in Patients with Atrial Fibrillation: A Pathway to Better Medical Education"

_ijerph, 2020, doi:10.3390/ijerph17114008_

Round 1

Reviewer 1 Report

Title/abstract:

  • Please, indicate the study design in the title or in the abstract

Introduction:

  • Page 2, line 48: Could authors specify (and reference) which guideline/s for atrial fibrillation (AF) treatment they refer to? They should also particularize if all AF guidelines make the same recommendations…

  • It would be clearer if the authors could summarize what are considered to be risk factors of stroke in the CHA2DS2-VASc score, as they do not include some common stroke risk factors

  • Page 2, Line 74: ” There remains a significant percentage of patients that are not receiving a correct antithrombotic treatment”…the authors should explain more and reference this sentence. In addition, if there is already evidence on this topic they should explain what the added value of their study is. Please explain the contribution of this study more clearly

Objectives:

  • Please, use the present tense to describe the objectives of the study.

  • It would be useful to add the setting of this study in the aim.

Methods:

  • Authors should specify the study design
  • Page 3, line 108: Please add the reference of the ESC clinical guideline.
  • Was the ESC guideline used as a comparison standard implanted at the hospital at the time of the study? If so, please add this information. If not, you should specify this point as a limitation.
  • Please, explain how the study size was arrived at
  • With respect to the data collection, there is evidence of underreporting of the clinical process in the medical records, a limitation of retrospective studies. The authors have recorded a large number of variables directly from the medical history as the only source, but some of the study variables are not usually collected routinely in medical records. Could the authors specify if they took this into account?

Results

  • Indicate number of participants with missing data for each variable of interest

Discussion

  • Authors should discuss the generalisability (external validity) of the study results
  • Authors should discuss limitations of the study.

CONCLUSION

  • Please, add the main result of your study: “more than 20% of patients with AF at risk for stroke do not receive the recommended OAC treatment”.

Minor point:

  • Page3, line 108: ESC instead of ECS

Author Response

Thank you very much for your valuable feedback, for your shared perspective and for the chance to improve our manuscript. With consideration, The Authors

Reviewer 2 Report

In this study the authors evaluated physicians adherence to antithrombotic drugs/oral anticoagulants use in patients with Atrialfibrilation. This manuscript is well written and methods and results are clearly discussed, except few minor comments below: In Abstract line 36: Result part is confusing. Total patients enrolled is 784, in which 617 with OAC and 167 without OAC. It is not 161 and Please check the percentage also. Did authors observed any gander effect on the results. Please discuss if any Similarly, race effect.

Author Response

Thank you very much for your valuable feedback and for the chance to improve our manuscript. Kind regards, The Authors

Reviewer 3 Report

This is a survey study investigating adherence of physicians to guidelines for anticoagulation prescription. The Authors included a high-risk population as seen by the high prevalence of heart failure and hypertension. Some points for the authors to consider:

  • Please add study design (retrospective) to the abstract.
  • In table 1, “No AC recommendation/ AC should be considered” is misleading
  • Factors included in the multivariable models should be clearly reported
  • I am not sure that this is the correct way to analyse the adjusted HASBLED as it should be considered as a time-dependent factor. Please consider removing it
  • It is interesting that bleeding risk is more important for physicians than cardiovascular comorbidity burden to decide on AC. It is also to note that paroxysmal AF, that means sinus rhythm is perceived as at lower risk of ischemic events, despite 77% of heart failure in this group.
  • Please specify on table 1 if liver and renal disease are defined according to the HASBLED or if other definitions are used.
  • The implications of this study should be improved. A holistic approach to patients with AF has been recently proposed by the ABC pathway, in which A represent appropriate antithrombotic treatment (Nat Rev Cardiol. 2017 Nov;14(11):627-628). This approach resulted in a significant reduction of ischemic complications (Mayo Clin Proc. 2019 Jul;94(7):1261-1267). The Authors should comment on this aspect.
  • The references from 28 to 35 seem to be excessive for the message which is not the main focus of this study.

Author Response

Thank you very much for your valuable feedback and your shared perspective. Your review gives us the chance to improve our manuscript. Kind regards, The Authors

Round 2

Reviewer 1 Report

The quality of the manuscript has improved after revision. Congratulations.